# Focusing: View-Consistent Sparse Voxels for Efficient 3D VAE Training

**Xuhui Chen** [* 1 2 3]   **Chao Long** [* 3]   **Fei Hou** [1 2]   **Dongbo Zhang** [3]   **Shaohui Jiao** [3]   **Wencheng Wang** [1 2]   **Ying He** [4]

## Abstract

High-fidelity 3D generation depends on 3D VAEs that can compress and reconstruct complex geometry at high resolution. Recent methods often convert raw meshes into signed distance fields (SDFs), but this preprocessing can be lossy, especially for open or non-watertight assets. Render-supervised VAEs such as TripoSF avoid this conversion by matching rendered depth and normal maps, yet render losses supervise only the visible geometry for each view, leaving many latent voxels weakly constrained while still incurring unnecessary decoding and attention costs. We introduce **Focusing**, a view-consistent sparse-voxel training scheme for efficient 3D VAEs. Given a training view, Focusing performs depth-driven voxel carving directly in the structured latent space: voxels inconsistent with the rendered depth are removed before decoding, allowing the decoder and attention layers to operate only on locally relevant geometry. This view-dependent sparsity reduces memory and computation while concentrating learning on surface regions that contribute to the render. To stabilize training across shapes and viewpoints, we further propose adaptive zooming, which adjusts camera intrinsics to keep the number of active voxels within a target range and strengths supervision for fine details. The VAE is trained with render-based depth, normal, mask, and perceptual losses, together with sparse-voxel total variation and a brief TSDF warm-up to improve convergence and suppress holes. Across standard reconstruction benchmarks, Focusing improves Chamfer Distance and F-score over strong baselines while substantially reducing video random access memory (VRAM) consumption, enabling $1024^3$-resolution VAE training with as little as 50 GB of VRAM. These results demonstrate that local, view-consistent sparsity is an effective path toward higher-resolution and more efficient 3D VAE training.

## 1. Introduction

3D generative AI has become an important research direction with broad applications in embodied AI, digital content creation, gaming, 3D modeling, and AR/XR. Compared with 2D image synthesis, however, 3D generation is substantially more challenging: models must handle higher-dimensional spatial signals, irregular geometry, complex topology, and diverse asset formats. As a result, producing high-fidelity 3D assets in a manner that is both accurate and computationally efficient remains difficult despite recent rapid progress.

Following the success of latent generative models in 2D, many 3D generation pipelines adopt a two-stage design. In the first stage, a 3D variational autoencoder (VAE) compresses an input representation, such as an SDF, point cloud, or voxel grid, into a compact latent space. In the second stage, a generative model, often a latent diffusion or rectified-flow model, operates in this reduced space to synthesize novel 3D assets. Within this pipeline, the VAE is a critical component: the fidelity and efficiency of its reconstruction directly determine the quality, scalability, and usability of the final generative model.

A central challenge is that 3D data lacks a universally accepted compact representation. Early 3D VAE designs such as VecSet (Zhang et al., 2023) encode input geometry into global latent tensors, but this representation introduces substantial redundancy because each feature is correlated with the entire shape. More recent methods, such as TREL-LIS (Xiang et al., 2025), address this issue with structured sparse latents, where each voxel stores local geometric information. This design improves locality, supports more structured latent representations, and enables local editing by manipulating specific voxels. However, efficiently training such high-resolution structured latents remains challenging, especially when the supervision signal does not constrain

---
[*]Equal contribution  [1]Key Laboratory of System Software (Chinese Academy of Sciences), Institute of Software, Chinese Academy of Sciences, China [2]University of Chinese Academy of Sciences, China [3]Bytedance, Beijing, China [4]College of Computing and Data Science, Nanyang Technological University, Singapore. Correspondence to: Fei Hou <houfei@ios.ac.cn>, Dongbo Zhang <zhangdongbo@bytedance.com>.

*Proceedings of the $43^{rd}$ International Conference on Machine Learning*, Seoul, South Korea. PMLR 306, 2026. Copyright 2026 by the author(s).

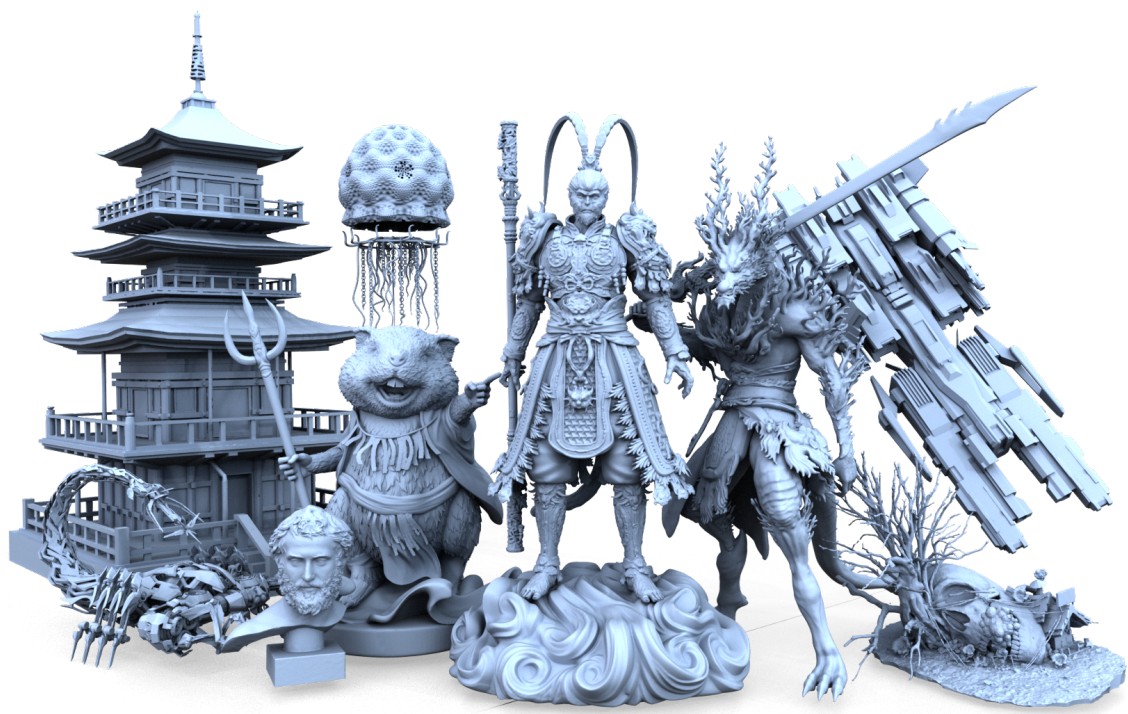

*Figure 1.* **Focusing** is an efficient and effective 3D VAE training framework, capable of generating high-fidelity 3D models at $1024^3$ resolution using less than 50GB of memory by using the render loss with an efficient voxel selection.

all voxels equally.

Sparc3D (Li et al., 2025) and Direct3D-S2 (Wu et al., 2025) extend this line by using SDFs for both input and output, thereby enforcing a unified modality. However, since most raw meshes are not watertight, these methods require lossy preprocessing steps to obtain SDFs. To overcome this limitation, rendering-based supervision is adopted by TripoSF (He et al., 2025), where the VAE is trained to match rendered depth and normal maps instead of precomputed SDF values. This avoids watertight conversion altogether, making the pipeline more flexible for open or complex training data. To further reduce computation, they adopt Frustum-aware Sectional Voxel Training, which prunes voxels outside the rendering frustum, thereby lowering training costs and enabling $1024^3$ upsampling. While such strategies reduce redundant computation, they also raise a fundamental question: *What is the minimum amount of computation required to render an image from a given viewpoint?*

TripoSF uses a point cloud as input and produces a mesh as output. This setup resembles the classical point cloud reconstruction pipeline, where high-quality meshes can be obtained from dense, oriented point clouds even without neural networks. Unlike TRELLIS, which requires global consistency to reproduce texture, a geometry-focused 3D VAE only needs local point distributions to recover surface

patches.

Motivated by this observation and the redundancy of existing approaches, we propose **Focusing**, a local 3D VAE training scheme built on a simple yet effective voxel carving. Following the TripoSF framework, we supervise the VAE using depth and normal maps rendered from ground-truth views. The key difference is that, before decoding, we perform depth-driven voxel carving directly in the structured latent space, as shown in Figure 2: voxels in the structured latent are compared with the rendered depth map, and only those consistent with the view are retained. This design removes most irrelevant computations before the attention and decoding stages, allowing the model to focus its capacity on fine-grained local geometry that actually contributes to the rendering loss.

To further improve detail capture and control memory usage, we introduce an adaptive zooming strategy. The number of visible voxels can vary significantly across objects and viewpoints, causing unstable VRAM consumption during training. Instead of relying only on near- or far-plane adjustment, we dynamically modify the camera intrinsics so that the number of activated voxels stays within a target range. This mechanism stabilizes training costs while also enabling zoomed-in supervision of local surface regions, which helps the model learn high-frequency geometric details. In addi-

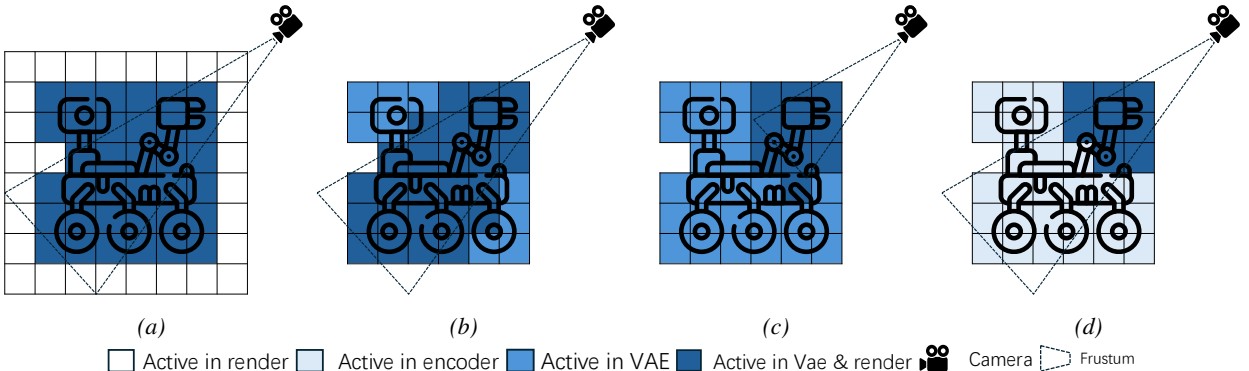

*Figure 2.* Conceptual illustration of sparse voxels in existing methods and in our approach. (a) TRELLIS encodes 3D models using sparse voxels and generates a mesh on a dense voxel grid for rendering. (b) TripoSF employs SparseFlex to reduce the number of voxels used for mesh extraction during rendering. (c) By adjusting the camera's far plane, TripoSF further reduces voxels that do not contribute to the final render. (d) Our method selects active voxels in the structured latent space based on the depth map. This strategy greatly reduces the computational overhead of both decoding and rendering, and it does not rely on the choice of the far plane.

tion, we use sparse-voxel total variation regularization to reduce small holes and a short TSDF warm-up to accelerate early convergence when render-based gradients are initially unavailable or unstable.

Together, depth-driven voxel carving and adaptive zooming enable efficient high-resolution VAE training while preserving reconstruction fidelity. Focusing substantially reduces unnecessary attention and decoding operations, lowers peak VRAM usage, and allows $1024^3$-resolution training on as little as 50 GB of VRAM. Experiments on standard reconstruction benchmarks show that Focusing improves geometric accuracy over strong baselines in terms of Chamfer Distance and F-score, while producing high-quality meshes with fine surface details. See Figure 1 for examples of high-resolution 3D models produced by our method.

## 2. Related Work

### 2.1. 3D Representations

**Meshes.** Meshes are the most common and versatile representation for 3D assets, and they are directly applicable to many downstream applications, such as rendering, animation, and simulation. A line of work explores treating meshes as sequences and generating them with sequence-to-sequence models (Nash et al., 2020; Siddiqui et al., 2024; Hao et al., 2024; Gao et al., 2025; Lionar et al., 2025). These methods can produce meshes with stylized or artist-like qualities, particularly quad meshes, but they are typically constrained by sequence length, which limits resolution and geometric fidelity. Other approaches attempt to directly predict mesh topology and geometry from images (Wang et al., 2018) or point clouds (Hanocka et al., 2020). More recently, differentiable rendering pipelines have been employed to supervise mesh generation with 2D projections. For example, TripoSF (He et al., 2025) represents meshes

using SparseFlex within a differentiable rendering pipeline, enabling efficient training and supporting higher-resolution mesh generation compared to prior mesh-based methods.

**Point Clouds.** Point clouds, often obtained from scanning devices, have long served as input for 3D reconstruction algorithms. Their connectivity-free nature and flexibility make them appealing as neural network input (Qi et al., 2017a;b). Other approaches generate point clouds directly as output distributions (Achlioptas et al., 2018; Yang et al., 2019; Liu et al., 2021; Luo & Hu, 2021), which allows for flexible and high-precision modeling. However, converting point clouds into watertight meshes is both lossy and computationally expensive (Kazhdan et al., 2006; Kazhdan & Hoppe, 2013; Hou et al., 2022), which limits their use in many downstream tasks.

**Implicit Functions.** Implicit functions represent 3D geometry as level sets of continuous fields, such as the zero level set of a signed distance function or the $1/2$-level set of an occupancy field. This formulation is robust and naturally compatible with neural networks, which explains its popularity in both reconstruction and generation tasks (Park et al., 2019). Since implicit functions do not directly produce meshes, an additional extraction algorithm (e.g., marching cubes (Lorensen & Cline, 1987) or its variants) is needed to recover surfaces. However, SDF and occupancy based approaches assume watertight geometry, which limits their ability to work with open models. To relax this constraint, UDFs have been proposed (Chibane et al., 2020), but existing UDF extraction methods often suffer from inefficiency and instability (Zhou et al., 2022; Hou et al., 2023; Ren et al., 2023). A common workaround is to preprocess training data by inflating meshes into watertight versions, yet this conversion can introduce artifacts and reduce geometric fidelity. For example, Sparc3D (Li et al., 2025) adopts

a flood-filling and deformation-based repair pipeline that inevitably leads to losing fine-scale details.

## 2.2. 3D VAE Architectures

Inspired by the success of 2D diffusion models (Rombach et al., 2022), most current 3D generative models follow a two-stage framework: a 3D VAE first compresses the input representation into a compact latent space, and a 3D diffusion model then operates in this reduced space to generate novel shapes. This design allows high-resolution synthesis while keeping diffusion tractable. Unlike 2D AIGC, however, there is no universally accepted compact representation for 3D data. The vast majority of available assets are stored as non-compact meshes, which are not ideal for direct neural network training due to irregular connectivity and variable topology.

3DShape2VecSet (Zhang et al., 2023) adopts a VAE to learn a compact vector-set (vecset) representation from point clouds sampled on meshes. The decoder is trained by supervising SDF values at query points, providing implicit geometric supervision during reconstruction. Clay (Zhang et al., 2024) extends this approach to large-scale datasets and introduces an inflation-based preprocessing pipeline to enforce watertight meshes. Dora (Chen et al., 2025) and Huanyuan2 (Tencent Hunyuan3D Team, 2025) enhance this framework with importance sampling, which improves the ability to capture sharp features. Hi3DGen (Ye et al., 2025) further incorporates normal-map supervision to boost the fidelity of surface detail reconstruction. Despite these advances, vecset representations suffer from significant information redundancy: each feature is correlated with the entire 3D model, making training inefficient and hindering scalability.

Both XCube (Ren et al., 2024) and TRELLIS (Xiang et al., 2025) replace the global vecset with sparse voxels. In this formulation, each voxel encodes local geometric information, leading to higher-quality reconstructions and more structured latent representations. TRELLIS (Xiang et al., 2025) further demonstrates that selectively replacing certain voxels enables flexible local 3D editing. Direct3D-S2 (Wu et al., 2025) introduces Spatial Sparse Attention to restrict computations to local neighborhoods, thereby reducing overhead during the diffusion stage. Sparc3D (Li et al., 2025) proposes a new preprocessing strategy to improve geometric fidelity. TripoSF (He et al., 2025) shifts away from SDF-based supervision and instead employs a rendering loss that compares predicted depth and normal maps with ground truth. This approach avoids the accuracy degradation caused by lossy watertight conversion and allows the VAE to handle both internal structures and open boundaries. However, render-based supervision provides weaker constraints than SDF supervision, since only voxels contributing to visible surfaces are directly trained. As a result, many latent voxels remain under-regularized, which can limit reconstruction accuracy and consistency.

## 3. Method

### 3.1. Preliminaries

TripoSF (He et al., 2025) introduced SparseFlex, a sparse version of FlexibleCubes (Shen et al., 2023), to train 3D AIGC models. SparseFlex is defined by a set of voxels $\mathcal{V}$. Each voxel $v_i \in \mathcal{V}$ contains both its spatial location $(x_i, y_i, z_i)$ (the 3D coordinates of its center) and feature information. Let the number of voxels be $N_v$ and the number of corresponding corners be $N_c$. The features include the SDF values $\{s_j \mid 0 \le j < N_c\}$ and deformations $\{\delta_j \mid 0 \le j < N_c\}$ for the voxel's eight corners. In practice, the features for each corner are obtained by averaging the values from surrounding relevant voxels. Additionally, the features also contain the interpolation weights $\{\alpha_i \in \mathbb{R}_{>0}^8, \beta_i \in \mathbb{R}_{>0}^{12} \mid 0 \le i < N_v\}$ per voxel for Dual Marching Cubes (DMC) (Nielson, 2004). Formally, the SparseFlex representation, $S$, is defined as:

$$\mathcal{S} = (\mathcal{V}, \mathcal{F}_c, \mathcal{F}_v), \quad \mathcal{F}_c = \{s_j, \delta_j\}, \quad \mathcal{F}_v = \{\alpha_i, \beta_i\}, \tag{1}$$

where $\mathcal{F}_c$ contains the SDF values and deformations at the corner grids, and $\mathcal{F}_v$ contains the interpolation weights for each voxel.

SparseFlex significantly reduces memory consumption and cuts down on unnecessary computational costs. Moreover, by supervising with a rendering loss instead of direct SDF values, TripoSF avoids the lossy watertight mesh conversion process. To further reduce computational overhead, TripoSF introduced Frustum-aware Sectional Voxel Training. This method applies SparseFlex to extract meshes only from voxels that are within the current camera's Normalized Device Coordinates (NDC) space. By adjusting the camera's intrinsic parameters, this cropping operation also enables the learning of a model's internal structure.

TripoSF then trains a VAE with the following losses:

$$\mathcal{L} = \lambda_1 \mathcal{L}_{\text{render}} + \lambda_2 \mathcal{L}_{\text{occ}} + \lambda_3 \mathcal{L}_{\text{KL}} + \lambda_4 \mathcal{L}_{\text{flex}} \tag{2}$$

$\mathcal{L}_{\text{render}}$ is the rendering supervision loss, including the following items:

$$\mathcal{L}_{\text{render}} = \lambda_d \mathcal{L}_d + \lambda_n \mathcal{L}_n + \lambda_m \mathcal{L}_m + \lambda_{ss} \mathcal{L}_{ss} + \lambda_{lp} \mathcal{L}_{lp} \tag{3}$$

where $\mathcal{L}_d$, $\mathcal{L}_n$, and $\mathcal{L}_m$ denote the $L_1$ loss for depth maps, normal maps, and mask maps, respectively. $\mathcal{L}_{ss}$ and $\mathcal{L}_{lp}$

denote SSIM loss and LPIPS loss, and are only applied to normal maps. TripoSF culls voxels that are far from the surface during the upsampling process in the decoder. The $\mathcal{L}_{\text{occ}}$ loss is used to guide the self-pruning upsampling module employed by TripoSF to accurately remove these distant voxels. Although this approach can reduce the number of voxels, it also creates more holes and makes convergence more difficult. $\mathcal{L}_{KL}$ is the KL divergence between the learned latent distribution and a standard normal prior, which helps to regularize the latent space. $\mathcal{L}_{\text{flex}}$ is the regularization term from Flexicubes that promotes smooth SDF values.

While SparseFlex avoids the accuracy loss from computing SDFs, its training still faces several issues:

- **Irrelevant voxels in the NDC space.** A significant number of invisible voxels participate in mesh extraction and require large VRAM, which significantly influences sparse voxel resolution scaling up. Although controlling the near and far planes can help, this approach is often suboptimal. To address this, we propose a plug-in-and-play visibility-based voxel carving strategy in Section 3.2, which more efficiently decreases the number of active voxels. We also show that this pruning can be performed in the latent space, which greatly reduces the decoder's workload on irrelevant voxels.

- **Rendering blurriness.** Methods based on rendering loss can suffer from blurriness due to the resolution of the rendered image. Capturing accurate detail often requires a higher resolution or a closer camera view, with the former significantly increasing computational cost. In Section 3.3, we introduce an adaptive camera adjustment strategy based on adaptive zooming. By controlling the number of voxels to be activated within the view, we can flexibly adjust camera parameters to better supervise the model's fine details.

- **Generating unnecessary holes.** Unlike direct SDF supervision, which provides a strict signal for maintaining watertightness, rendering loss struggles to provide effective supervision for small holes. In Section 3.4, we introduce our VAE framework and propose a new regularization loss to reduce these holes.

### 3.2. Depth-based Voxel Carving

Our key insight is that the local surface is determined by local points, independent of distant points. As shown in Figure 3, selecting only a subset of voxels in the latent space and feeding them to the decoder can still generate a locally complete mesh, except for some jagged noise at the boundary. Because a rendering-loss-based VAE does not depend on a globally complete mesh, we can reduce the number of voxels used by the network without changing the rendering result, provided that we align the input camera with the filtered voxels.

Specifically, given a camera with extrinsic $\pi$, intrinsics $K$, and the near $(n)$ and far $(f)$ clipping planes of the viewing frustum, we compute the Model-View-Projection matrix to render the depth map $D$ from the input raw mesh. For each voxel $z_i$ in the latent code $\mathcal{Z}$, we also project its center to obtain the projected point $z_i^p = \{u_i^p, v_i^p, d_i^p\}$ in the image space and regard $d_i^p$ as the depth of each voxel $z_i$. We assume that the camera is directed toward the negative $z$-axis, implying that all depth values are negative. Using a threshold $r$, we only retain voxels satisfying $d_i^p \leq D(u_i^p, v_i^p) - r$, effectively discarding voxels that are far from visible surfaces. The remaining voxels are denoted by $\mathcal{Z}_{\text{carve}}^p$. To enhance robustness at the image boundary, the $3 \times 3 \times 3$ neighborhood of any remaining voxel is also retained.

### 3.3. Adaptive Zooming

The large changes in how many voxels are visible cause significant VRAM usage fluctuations during training. TREL-LIS addresses this by removing voxels exceeding a fixed quota, but this can impact the final rendering if essential voxels are lost. TripoSF utilizes a visibility ratio $\alpha$, to control the number of active voxels, primarily by adjusting the near and far planes. This approach relies on the camera being preset close to the object's surface to capture key details.

In contrast, since we have already removed most invisible voxels, adjusting the far plane has little influence on the number of voxels. We introduce a zooming-based method for adjusting voxel count. This approach effectively controls the number of active voxels while enabling flexible zoom-in operations to capture finer model details.

We randomly select a visibility ratio $\alpha \in [\alpha_{\min}, \alpha_{\max}]$ and retain only $\alpha N$ voxels to capture various levels of geometric structure, where $N$ is the number of voxels in $\mathcal{Z}_{\text{carve}}^p$. To limit the maximum number of voxels and avoid having too few, $\alpha N$ is clamped between $N_{\min}$ and $N_{\max}$. Specifically, we randomly select a voxel to serve as a seed, and search for the $\alpha N$ closest voxels to this seed in the camera space to obtain the cropped voxel set $\mathcal{Z}_{\text{crop}}^p$ from $\mathcal{Z}_{\text{carve}}^p$. The camera space coordinates of these voxels are used to calculate the initial bounding box $(x_{\min}^n, x_{\max}^n, y_{\min}^n, y_{\max}^n)$. Since this bounding box contains greater than or equal to $\alpha N$ voxels, we subsequently use a binary search method to update our bounding box until the difference between the number of voxels inside the bounding box and the target number is less than $5\%$. We then adjust the perspective matrix $P$ using the new image bounding box as follows:

Rendering Loss

Point cloud based

Voxelization

Encoder

Voxel carving

Decoder

Voxel Upsampling

Sparse Flexicube

*Figure 3.* Overview of our framework. Starting from the input mesh, we first voxelize it and aggregate local features from sampled point clouds to form input voxels. A sparse transformer encoder-decoder then compresses these structured features into a latent space, followed by an upsampling module to increase resolution. In the latent space, we perform visibility-based voxel carving to retain only view-consistent voxels that contribute to the visible mesh in the rendered image. The refined structured features are decoded into sparse flexible cube representation for final mesh extraction. Supervision is provided by a rendering loss, which compares the rendered images of the input and reconstructed meshes.

$$P_{\text{new}} = \begin{pmatrix} \frac{2s}{x^n_{\max}-x^n_{\min}} & 0 & 0 & -\frac{x^n_{\max}+x^n_{\min}}{x^n_{\max}-x^n_{\min}} \\ 0 & \frac{2s}{y^n_{\max}-y^n_{\min}} & 0 & -\frac{y^n_{\max}+y^n_{\min}}{y^n_{\max}-y^n_{\min}} \\ 0 & 0 & 1 & 0 \\ 0 & 0 & 0 & 1 \end{pmatrix} P \tag{4}$$

where $s$ is the scaling rate used to prevent jagged noise from appearing at the boundary of the image.

### 3.4. VAE Structure

Following TripoSF (He et al., 2025), we use a variational autoencoder that compresses the input oriented dense point cloud into a sparse latent code $\mathcal{Z}$ without relying on computationally expensive global attentions. A frozen Point-Net (Qi et al., 2017a) adapted from TripoSF is used to aggregate local geometric features within each voxel. A sparse transformer then utilizes the shifted window attention proposed by TRELLIS (Xiang et al., 2025) to learn relationships between voxels. This process outputs a fused structured latent feature $\mathcal{Z}$, where each voxel possesses local geometric information. Specifically, the initial feature dimension of each latent voxel in the latent space is 8. Within the 3D sparse decoder, this feature is expanded to a 64-dimensional hidden representation and is subsequently mapped by the final linear projection layer into a final 53-dimensional vector. The voxel carving and adaptive zooming are then used to filter the voxels in $\mathcal{Z}$ to attain $\mathcal{Z}^p_{\text{crop}}$. The decoder takes $\mathcal{Z}^p_{\text{crop}}$ as input and uses a series of transformer layers to generate the final output. To support high-resolution output, two self-pruning upsampling modules are then employed to obtain a $2\times$ upsampled result from the initial output, ultimately yielding a $4\times$ resolution output. We only use $\mathcal{L}_{\text{occ}}$ to supervise the self-pruning block and do not use the output of the self-pruning block, $O_{\text{pred}}$,

to filter voxels during training, in order to avoid unnecessary holes in the rendered images that negatively impact the rendering loss.

In our framework, the parameters of the output layer are initialized to zero. Consequently, the initial SDF values predicted by the VAE are exactly zero, making it impossible to extract a valid mesh for gradient backpropagation via rendering loss. To start and guide the convergence, we supervise the SDF $s_j$ of the VAE output using the TSDF $s^{\text{raw}}_j$ of the input raw mesh during the very early stages of training.

$$\mathcal{L}_{\text{tsdf}} = \sum_{j=1}^{N_c} \|s_j - s^{\text{raw}}_j\| \tag{5}$$

To reduce the generation of holes, we use the Total Variation (TV) loss on sparse voxels to suppress the differences in SDF values between adjacent corners:

$$\mathcal{L}_{\text{tv}} = \sum_{v \in \mathcal{V}} \sqrt{\Delta^2_x(v) + \Delta^2_y(v) + \Delta^2_z(v)} \tag{6}$$

where $\Delta^2_x(v)$ denotes the squared difference of SDF values between voxel $v := (i, j, k)$ and its adjacent voxel at $(i + 1, j, k)$, which can be analogously extended to $\Delta^2_y(v)$ and $\Delta^2_z(v)$. We apply this Total Variation (TV) term directly to the SDF grid, denoted by $\mathcal{L}_{\text{tv}}$, to encourage a continuous and compact geometry.

The overall loss function is:

$$\mathcal{L} = \lambda_1 \mathcal{L}_{\text{render}} + \lambda_2 \mathcal{L}_{\text{occ}} + \lambda_3 \mathcal{L}_{\text{KL}} + \lambda_4 \mathcal{L}_{\text{flex}} + \lambda_5 \mathcal{L}_{\text{tv}} + \lambda_6 \mathcal{L}_{\text{tsdf}} \tag{7}$$

### 3.5. Rectified Flow based Image-to-3D Generation

Drawing inspiration from TRELLIS (Xiang et al., 2025) and TripoSF (He et al., 2025), we employ a two-stage rectified-

flow generation model, which consists of a sparse structure flow model and a structured latents flow model.

**Structured latent flow model**. Based on our proposed VAE, we first encode the sampled point cloud and its corresponding sparse structure into the latent space. Similar to the sparse structure flow model, we use the DINO (Siméoni et al., 2025) features of the condition image as the condition to train the DiT via cross-attention. We then use a rectified flow model for denoising. Finally, the denoised latents are decoded into a 3D shape by the VAE's decoder.

## 4. Experimental Results

### 4.1. Setup

**Implementation Details.** Following TRELLIS (Xiang et al., 2025), we train both the VAE and its latent flow model on 183K high-quality assets from Objaverse-XL (Deitke et al., 2023). We employ a progressive training scheme for our VAE. The $512^3$ resolution VAE runs on 32 A800 GPUs (batch size 32) with AdamW (initial LR 1 X $10^{-4}$, weight decay 0.01) for two days. A cosine annealing learning rate schedule with 40K steps is used to adjust the learning rate. We then train the $1024^3$ resolution VAE with 32 A100 GPUs using the same settings.

**Hyperparameter Settings.** We use the same weight configuration as TRELLIS to train our VAE. For our two newly introduced losses, we set $\lambda_5 = 0.001$ and $\lambda_6 = 1$. The $\mathcal{L}_{tsdf}$ is only used for the first 12K steps. We set the threshold $r = 2/\text{Res}$ to carve voxels. We use a resolution of $518^2$ to render images and use $s = 1.1$ to calculate the new perspective matrix. For zooming, we set $\alpha_{\min} = 0.1$ and $\alpha_{\max} = 0.3$ to train our model. We set $N_{\min} = 8192$ and $N_{\max} = 15360$ to limit the number of voxels.

**Camera Viewpoint Sampling.** To completely eliminate the risk of under-representing any surface region, we designed a "Hybrid of Uniform Sampling and Stratified Zooming" strategy for training data preparation (300 views in total). First, we generate 200 views using a Fibonacci spherical lattice, which mathematically guarantees a nearly perfect isotropic uniform distribution. These views use fixed frustum: 100 of them use a far scale (Radius 2.0, FOV $40°$), and the other 100 use a medium scale (Radius 1.5, FOV $40°$). Second, the remaining 100 views are generated using a Hammersley sequence with a random global phase offset. For these views, aggressive adaptive zooming is enabled (the radius is randomly sampled from $[0.8, 2.2]$ and FOV from $[20°, 70°]$) to act as stochastic data augmentation, forcing the latent code to learn multi-scale, high-frequency local details.

**Inference Details.** Unlike the training phase, we do not employ the view-dependent carving strategy during inference.

*Table 1.* Comparison of TRELLIS, Dora, Direct3D-S2, TripoSF, and our method in terms of input formats, preprocessing needs, latent feature representations, texture dependencies, and maximum supported resolution. IF: Image Feature; Raw: Raw Mesh (possible non-watertight); WT: Watertight mesh; SV: Sparse Voxel.

| Method | Input | Prep. | Feature | Texture | Max Res. |
|---|---|---|---|---|---|
| TRELLIS | IF | No | SV | Yes | 256 |
| Dora | WT | Yes | VecSet | No | 256 |
| D3D-S2 | WT | Yes | SV | No | 1024 |
| TripoSF | Raw | No | SV | No | 1024 |
| Ours | Raw | No | SV | No | 1024 |

Instead, the decoder processes the complete set of voxels to reconstruct the full geometry. To effectively control peak VRAM usage and accommodate the large number of voxels at inference time, we decode the latent codes by splitting them into multiple chunks.

*Table 2.* We sample 100K point cloud to measure the Chamfer Distance and F-score on the Dora Benchmark (Chen et al., 2025) in different geometry details Levels. Low Chamfer Distance ensures overall shape fidelity, while high F-score ensures local details are accurately covered within an acceptable error radius.

| | Chamfer Distance ($10^{-5}$) | | | | | | | |
|---|---|---|---|---|---|---|---|---|
| | L1 | | L2 | | L3 | | L4 | |
| | Mean | Std | Mean | Std | Mean | Std | Mean | Std |
| TripoSF$_{512}$ | 1.382 | 0.985 | 1.600 | 1.189 | 2.184 | 1.368 | 3.107 | 2.126 |
| Ours$_{512}$ | 1.353 | 0.995 | 1.513 | 1.008 | 2.116 | 1.351 | 2.806 | 1.771 |
| TripoSF$_{1024}$ | 1.315 | 0.937 | 1.456 | 0.936 | 2.007 | 1.197 | 2.431 | 1.390 |
| Ours$_{1024}$ | 1.294 | 0.886 | 1.429 | 0.873 | 1.901 | 1.011 | 2.264 | 1.055 |

| | F-score | | | | | | | |
|---|---|---|---|---|---|---|---|---|
| | L1 | | L2 | | L3 | | L4 | |
| | Mean | Std | Mean | Std | Mean | Std | Mean | Std |
| TripoSF$_{512}$ | 0.951 | 0.078 | 0.939 | 0.089 | 0.892 | 0.115 | 0.831 | 0.146 |
| Ours$_{512}$ | 0.953 | 0.080 | 0.945 | 0.086 | 0.897 | 0.117 | 0.841 | 0.147 |
| TripoSF$_{1024}$ | 0.957 | 0.074 | 0.951 | 0.078 | 0.907 | 0.106 | 0.873 | 0.121 |
| Ours$_{1024}$ | 0.958 | 0.070 | 0.953 | 0.073 | 0.916 | 0.091 | 0.886 | 0.097 |

*Table 3.* Comparison of training efficiency with $512^3$ resolution in A100 GPU.

| Method | Training Speed (steps/h) | GPU Memory Peak (GB) |
|---|---|---|
| w/o Carving & Zooming | OOM | OOM |
| w/o Zooming | 510 | 78 |
| Ours ($\alpha_{min} = 0.6$) | 657 | 67 |
| Ours ($\alpha_{min} = 0.3$) | 892 | 56 |
| Ours ($\alpha_{min} = 0.15$) | 1003 | 49 |

GT      TripoSF$_{512}$      TripoSF$_{1024}$      Ours$_{512}$      Ours$_{1024}$

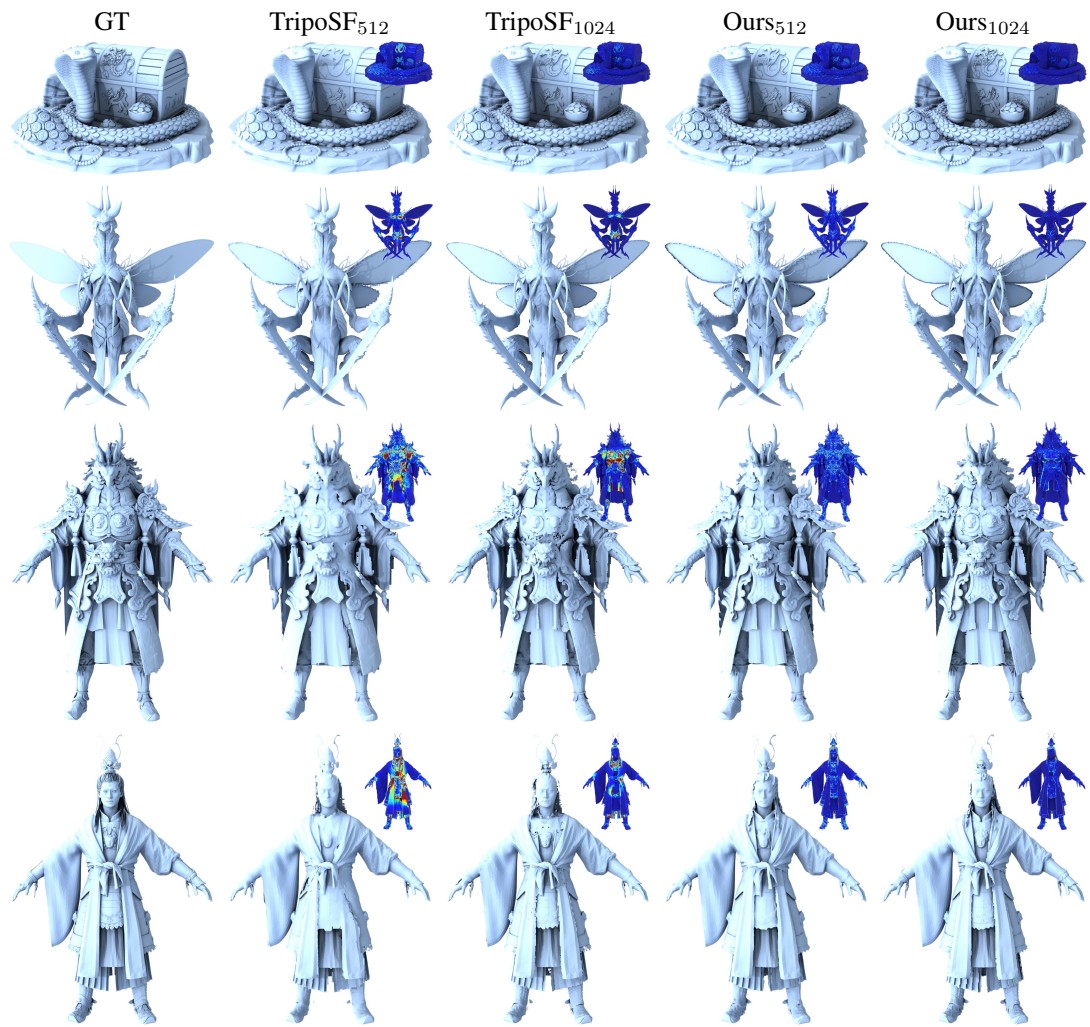

*Figure 4.* Qualitative comparison of VAE reconstruction between ours and TripoSF with different resolutions. Our approach demonstrates superior performance in reconstructing geometry details.

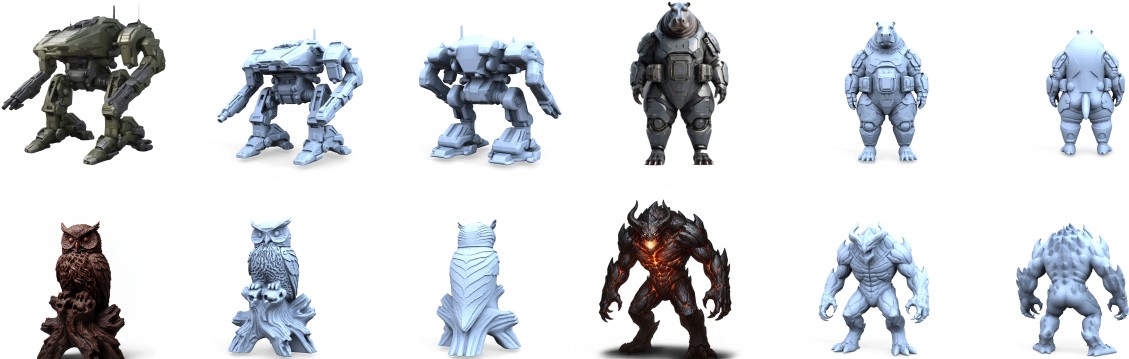

*Figure 5.* Single image-to-3D generations from AI generated images which are collected from Geminis or Dora.

## 4.2. VAE Reconstruction Evaluation

As shown in Table 1, although Dora-VAE, TRELLIS, and Direct3D-S2 all provide VAE weights, they can only reconstruct watertight models. Therefore, it is unfair to directly compare them with our method on (possibly non-watertight) raw meshes. We specifically show the comparison with

GT      Ours$_{256}$      Ours$_{512}$      Ours$_{1024}$      Ours$_{1536}$

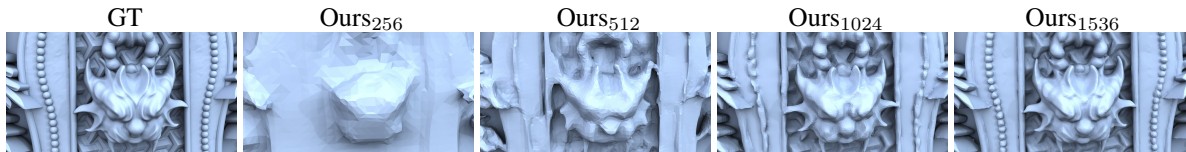

*Figure 6.* We use the $1024^3$ resolution checkpoint to test the reconstruction results at different resolutions. Our method demonstrates generalization capabilities across different resolutions.

these methods in Appendix A. Here, we primarily compare with TripoSF, as currently only TripoSF supports raw mesh reconstruction. Our method demonstrates superior VAE reconstruction performance, with quantitative results detailed in Table 2. Our model outperforms TripoSF at the same resolution in terms of the L2 norm of Chamfer Distance (CD) and F-score with a threshold 0.005. Since even the Level 4 data in the Dora Benchmarks lack sufficient details, we selected several detail-rich models from online websites to further test the detail reconstruction ability of our method, as shown in Figure 4. Our method outperforms TripoSF for detail reconstruction. In certain high-fidelity cases, our 512-resolution output recovers geometric details even more accurately than TripoSF's 1024-resolution result, as illustrated in Figure 4.

### 4.3. Image-to-3D Generation

We provide the image-to-3D generation results to verify that the latent space of our VAE complies with the KL divergence constraint and can be applied to 3D AIGC tasks. We do not claim that our rectified flow model is state-of-the-art, as we primarily focus on introducing a new training strategy for 3D VAEs. Nevertheless, visualizations in Figure 5, including image-to-3D results from AI generated images, highlight the generalization and effectiveness of our method. The generated 3D shapes maintain sharp edges and rich details while exhibiting high fidelity to the corresponding input images.

### 4.4. Ablation Studies

**Depth-based voxel carving and adaptive zooming.** In this paper, we employ depth-based carving and adaptive zooming to replace the Frustum-aware Sectional method used by TripoSF (He et al., 2025). This substitution allows for a more reasonable voxel cropping operation. We further find that this cropping operation can be directly applied in the latent space, which further reduces the computational cost. We present the impact of each component on the training speed and memory footprint in Table 3. It demonstrates that performing voxel carving based on visibility in the latent space significantly reduces training costs. Simultaneously, we can further adjust the computational cost by tuning $\alpha_{max}$. However, considering that the encoder still needs to encode global information, an excessively low $\alpha_{max}$ does not pro-

vide a linear reduction in cost. Furthermore, because the encoder still encodes global information, our method cannot be directly scaled up to train at $1536^3$ resolution. We also provide qualitative ablations regarding the choice of the carving threshold $r$ in Figure 9 of the Appendix, illustrating how different threshold values affect the preservation of geometry.

**VAE resolution.** Higher resolution always leads to better VAE reconstruction quality, as shown in Table 2. We additionally evaluated reconstruction performance across multiple resolutions using the VAE model trained at a $1024^3$ resolution, as shown in Figure 6. We found that our model has the ability to complete $1536^3$ resolution reconstruction without being trained at the corresponding resolution. At the $1536^3$ resolution, chain-beads that were previously not captured were successfully reconstructed.

## 5. Conclusions & Future Directions

We present Focusing, an efficient voxel-carving scheme for 3D VAE training in 3D model generation. By using only view-consistent voxels in the structured latent space for decoding, our method eliminates redundant computation and enables high-resolution, detail-preserving VAE training while lowering memory usage. Experiments show that Focusing improves both reconstruction accuracy and training efficiency over strong baselines.

One current limitation is that our pipeline uses point-cloud features as input and still depends on point normals to predict the final surface. For very thin structures, such as the hair-like geometry in the last example of Figure 4, locally consistent normals are difficult to estimate, and fine details may be missed.

In future work, we plan to explore more robust input features, such as multi-view image features, local surface descriptors, or learned orientation-free geometric features, to more effectively capture thin and sub-voxel structures. Another promising direction is to extend view-consistent voxel selection beyond VAE reconstruction to the full 3D generation pipeline. For example, the same focusing principle could be integrated into latent diffusion or rectified-flow models, allowing generative models to allocate more capacity to geometrically important regions while avoiding unnecessary computation in empty or invisible spaces.

# Acknowledgment

This work is partially supported by the Research Projects of ISCAS (ISCAS-JCMS-202303, ISCAS-ZD-202401, ISCAS-JCZD-202402 and ISCAS-JCMS-202403), and the Ministry of Education, Singapore, under its Academic Research Fund Grant RT19/22.

# Impact Statement

This paper presents work whose goal is to advance the field of Machine Learning. There are many potential societal consequences of our work, none of which we feel must be specifically highlighted here.

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

## A. Comparison with More Methods

We sample 100K points to measure the Chamfer Distance (CD) and F-score on the Dora Benchmark (Chen et al., 2025) for different levels of geometric detail. The comparisons are performed separately within each group classified by input type, as reported in Table 4.

**Image features.** TRELLIS (Xiang et al., 2025) uses image features as input, and thus can only achieve better reconstruction for the visible regions. Furthermore, the mesh extraction process in TRELLIS requires a dense grid, where the SDF values of inactive voxels are filled with 1.0. Since the valid sparse predictions contain negative SDF values indicating the inner space of the geometry, the abrupt transition from these negative values to the padded value of 1 forces the marching cubes algorithm to extract artificial surfaces. As a result, this operation generates a redundant layer of faces on the inner side of the object, as shown in Figure 8. Although TRELLIS proposed a post-processing method to remove these internal faces, it can result in erroneous removals that may, in fact, degrade the metrics. For fair comparisons, we use TRELLIS's visibility-based point cloud sampling results to calculate the CD and F-score.

**Watertight mesh.** Sparc3D (Li et al., 2025), Dora (Chen et al., 2025), and Direct3D-S2 (Wu et al., 2025) all require the input mesh to be a watertight mesh to compute the SDF. Dora uses the $\epsilon$-level set of the raw mesh as the watertight representation of the original model. Sparc3D proposed a method based on flood fill and optimization to further reduce the loss of precision caused by this conversion, but this may introduce self-intersections in the output. Since the code of Sparc3D is not open source, we use Dora's method for data processing: we set $\epsilon = 2/\text{Res}$ to extract the surface. This processed data is then used as the GT mesh and input to each method to obtain the reconstruction results. Since the $\epsilon$-level set method results in the loss of object surface details, and this loss is particularly severe at lower resolutions (higher $\epsilon$), our method achieves the best results only at Level 4 on the Watertight remesh results. Furthermore, at Level 1, the low-resolution results appear better than the high resolution ones, because there are so few fine details that would benefit from being captured at higher resolution.

**Raw mesh.** Evaluating methods directly on raw meshes most accurately represents their capability for geometric reconstruction, since some of the raw meshes are not watertight. However, as mentioned earlier, methods other than ours and TripoSF (He et al., 2025) cannot be directly applied to non-watertight meshes, which causes the metrics of these methods to fall significantly behind. To avoid unnecessary misunderstanding, we mark these methods with an asterisk ($*$). Although we do not consider internal viewpoints during VAE training, at inference time we retain all voxels in the latent space, and the VAE still successfully produces objects with internal structures, as illustrated in Figure 7.

Exterior · Cut view

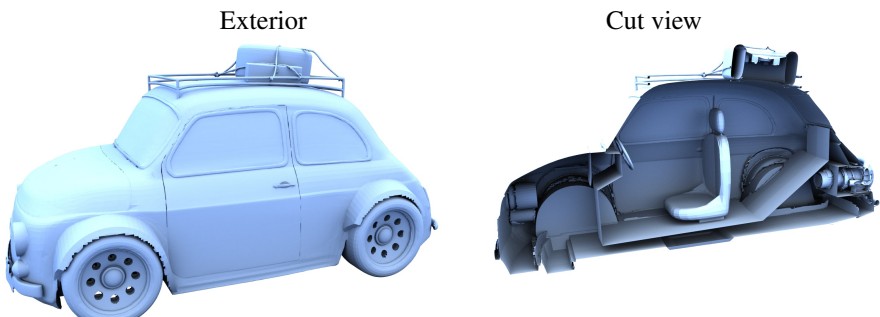

*Figure 7.* Our method is able to produce shapes with inner structures.

## B. Fail Case

3D VAEs that rely on voxelized point cloud features inherently struggle to capture thin structures whose thickness is smaller than one voxel. As a result, such structures may be omitted during reconstruction. As illustrated in Figure 10, both our method and TripoSF fail to recover extremely thin geometric elements. Handling these sub-voxel structures remains an open challenge for voxel-based representations and is an important direction for future work.

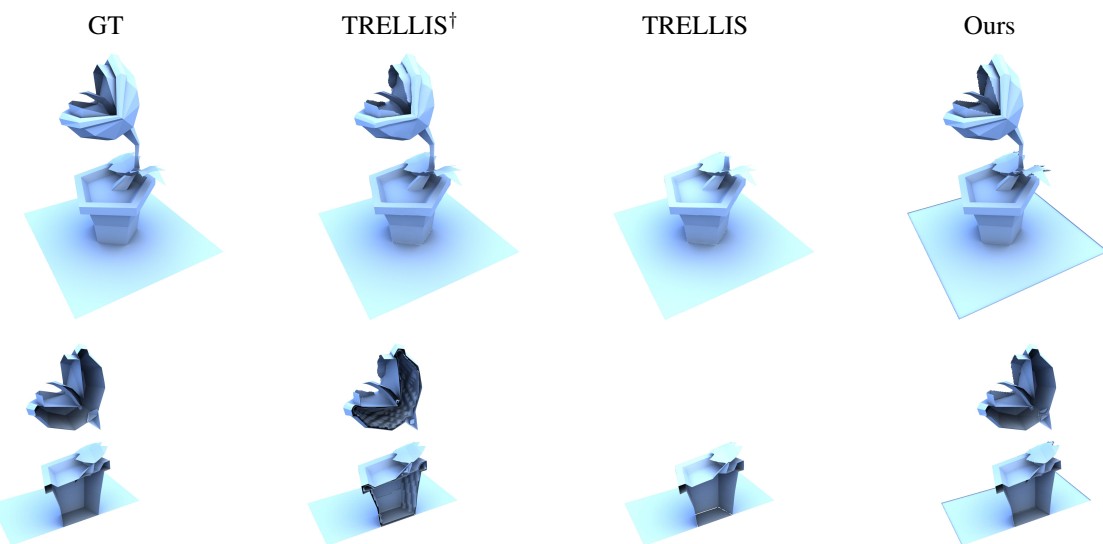

*Figure 8.* The inner redundant faces generated by TRELLIS ([†]without TRELLIS post-processing). Although TRELLIS uses a visible-based post-processing for removing redundant inner faces, some triangles might not be properly deleted or retained.

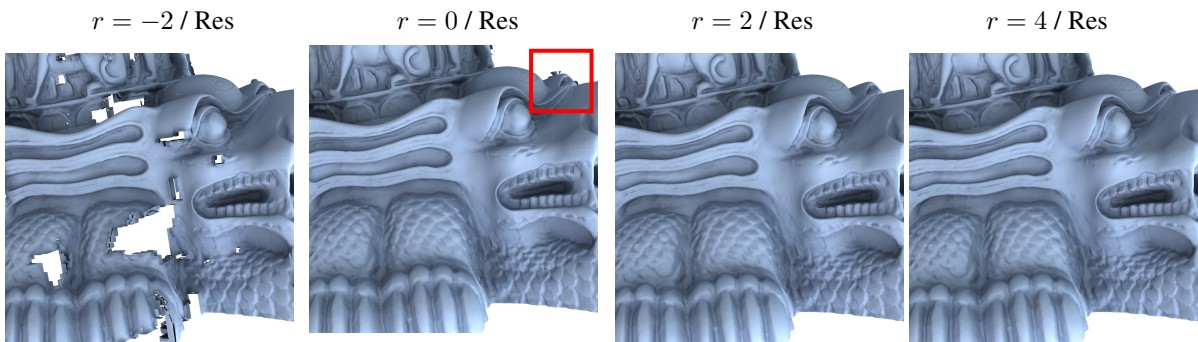

*Figure 9.* Since the centers of some voxels may lie behind the surface, we use $r = 2$ / Res as the depth-carving threshold to retain these voxels. A smaller $r$ can lead to geometric errors, while a larger $r$ has almost no effect on geometry but slightly increases the number of retained voxels.

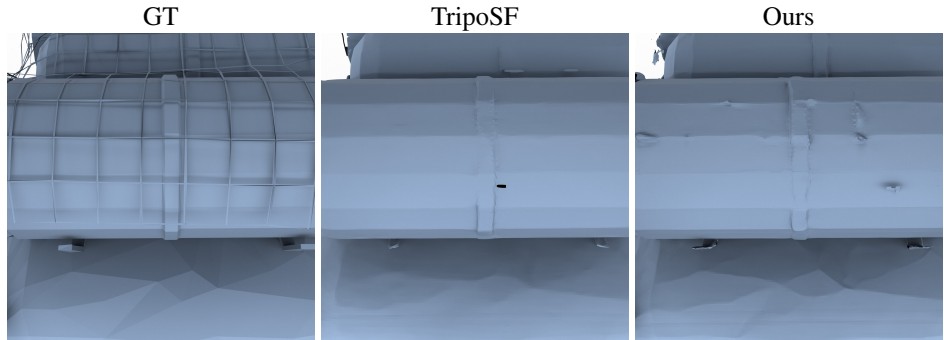

*Figure 10.* Fail case. Neither our method nor TripoSF can reconstruct extremely thin structures.

GT      Direct3D-S2$_{1024}$      TripoSF$_{1024}$      Ours$_{1024}$

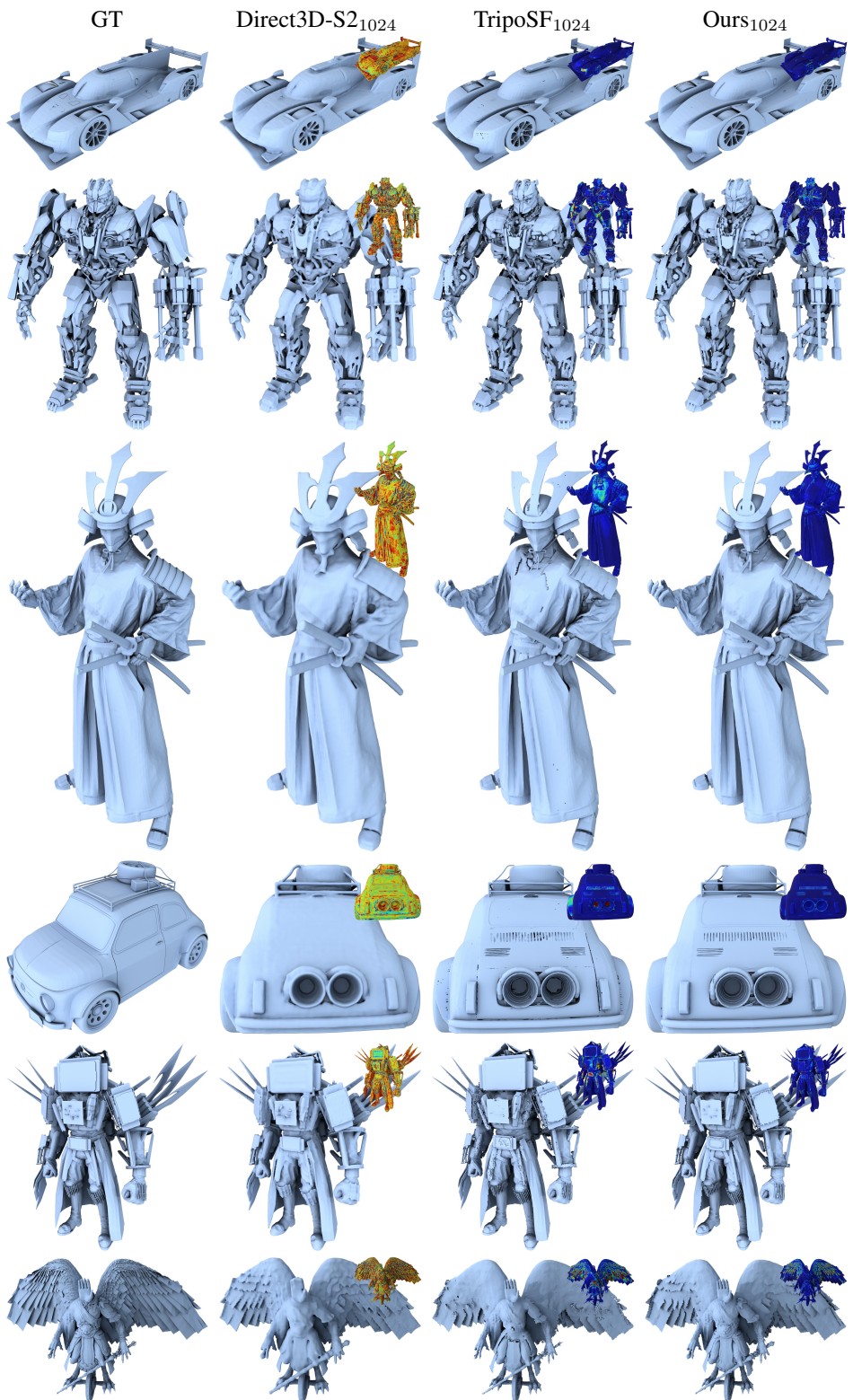

*Figure 11.* More qualitative comparison of VAE reconstruction between ours, TripoSF and Direct3D-S2 with $1024^3$ resolution. Our approach demonstrates superior performance in reconstructing geometry details.

*Table 4.* We sample 100K point cloud to measure the Chamfer Distance and F-score on the Dora Benchmark (Chen et al., 2025) in different geometry details Levels. Comparison is conducted within groups of various inputs (Image features, Watertight mesh processed by Dora and Raw mesh, respectively).

| | | Chamfer Distance $(10^{-5})\downarrow$ | | | | | | | |
| | | L1 | | L2 | | L3 | | L4 | |
| Input Type | Method | Mean | Std | Mean | Std | Mean | Std | Mean | Std |
|---|---|---|---|---|---|---|---|---|---|
| Image features | TRELLIS | 1.633 | 1.088 | 2.098 | 2.095 | 5.300 | 68.56 | 4.882 | 11.16 |
| | TRELLIS$^\dagger$ | 1.571 | 0.933 | 1.895 | 1.117 | 2.391 | 1.369 | 3.253 | 4.778 |
| Watertight mesh | Dora | 1.886 | 0.771 | 2.182 | 0.832 | 2.753 | 1.487 | 3.217 | 1.278 |
| | Direct3D-S2$_{1024}$ | 1.128 | 0.946 | 1.402 | 2.181 | 1.640 | 0.893 | 2.308 | 1.123 |
| | TripoSF$_{512}$ | 1.232 | 0.812 | 1.304 | 0.763 | 1.838 | 1.053 | 2.517 | 1.322 |
| | Ours$_{512}$ | 1.218 | 0.810 | 1.305 | 0.765 | 1.840 | 1.049 | 2.518 | 1.334 |
| | TripoSF$_{1024}$ | 1.243 | 0.825 | 1.321 | 0.741 | 1.844 | 1.007 | 2.426 | 1.184 |
| | Ours$_{1024}$ | 1.242 | 0.824 | 1.324 | 0.741 | 1.785 | 0.898 | 2.258 | 0.906 |
| Raw mesh | TRELLIS* | 266.8 | 2270 | 23.00 | 92.21 | 35.13 | 213.2 | 90.26 | 867.3 |
| | Dora* | 300.5 | 1808 | 225.7 | 1512 | 157.7 | 934.1 | 200.9 | 1447 |
| | Direct3D-S2$_{1024}$* | 335.3 | 1725 | 363.8 | 2161 | 415.2 | 3224 | 425.4 | 2336 |
| | TripoSF$_{512}$ | 1.382 | 0.985 | 1.600 | 1.189 | 2.184 | 1.368 | 3.107 | 2.126 |
| | Ours$_{512}$ | 1.353 | 0.995 | 1.513 | 1.008 | 2.116 | 1.351 | 2.806 | 1.771 |
| | TripoSF$_{1024}$ | 1.315 | 0.937 | 1.456 | 0.936 | 2.007 | 1.197 | 2.431 | 1.390 |
| | Ours$_{1024}$ | 1.294 | 0.886 | 1.429 | 0.873 | 1.901 | 1.011 | 2.264 | 1.055 |

| | | F-score $\uparrow$ | | | | | | | |
| | | L1 | | L2 | | L3 | | L4 | |
| Input Type | Method | Mean | Std | Mean | Std | Mean | Std | Mean | Std |
|---|---|---|---|---|---|---|---|---|---|
| Image features | TRELLIS | 0.946 | 0.075 | 0.928 | 0.078 | 0.897 | 0.080 | 0.861 | 0.074 |
| | TRELLIS$^\dagger$ | 0.945 | 0.076 | 0.928 | 0.078 | 0.896 | 0.081 | 0.862 | 0.074 |
| Watertight mesh | Dora | 0.959 | 0.069 | 0.937 | 0.081 | 0.887 | 0.108 | 0.838 | 0.117 |
| | Direct3D-S2$_{1024}$ | 0.974 | 0.048 | 0.969 | 0.055 | 0.941 | 0.074 | 0.887 | 0.106 |
| | TripoSF$_{512}$ | 0.964 | 0.061 | 0.963 | 0.061 | 0.920 | 0.094 | 0.862 | 0.121 |
| | Ours$_{512}$ | 0.965 | 0.061 | 0.963 | 0.061 | 0.920 | 0.093 | 0.862 | 0.122 |
| | TripoSF$_{1024}$ | 0.963 | 0.062 | 0.963 | 0.059 | 0.922 | 0.089 | 0.874 | 0.112 |
| | Ours$_{1024}$ | 0.963 | 0.062 | 0.962 | 0.059 | 0.927 | 0.079 | 0.889 | 0.087 |
| Raw mesh | TRELLIS* | 0.430 | 0.221 | 0.719 | 0.149 | 0.665 | 0.150 | 0.646 | 0.189 |
| | Dora* | 0.379 | 0.077 | 0.408 | 0.079 | 0.337 | 0.068 | 0.292 | 0.084 |
| | Direct3D-S2$_{1024}$* | 0.370 | 0.361 | 0.339 | 0.277 | 0.240 | 0.175 | 0.269 | 0.166 |
| | TripoSF$_{512}$ | 0.951 | 0.078 | 0.939 | 0.089 | 0.892 | 0.115 | 0.831 | 0.146 |
| | Ours$_{512}$ | 0.953 | 0.080 | 0.945 | 0.086 | 0.897 | 0.117 | 0.841 | 0.147 |
| | TripoSF$_{1024}$ | 0.957 | 0.074 | 0.951 | 0.078 | 0.907 | 0.106 | 0.873 | 0.121 |
| | Ours$_{1024}$ | 0.958 | 0.070 | 0.953 | 0.073 | 0.916 | 0.091 | 0.886 | 0.097 |

$^\dagger$ indicates without TRELLIS post-processing for removing redundant inner faces. * indicates these methods are not applicable to (possibly non-watertight) raw meshes, so the data are for reference only.

