# OpenReview forum: "Focusing: View-Consistent Sparse Voxels for Efficient 3D VAE Training"
_ICML.cc/2026/Conference — ICML 2026 regular_

### Official Review · Reviewer_jWX3 · 2026-03-08

**Soundness:** 3
**Presentation:** 2
**Significance:** 3
**Originality:** 3
**Overall Recommendation:** 4
**Confidence:** 3

**Summary:**

The paper presents a method to more efficiently encode and decode objects with 3D VAEs. Many recent works on 3D mesh generation like TRELLIS (Xiang et al., 2025) have popularized autoencoding shapes into latent voxels, but they remain memory intensive. To address this issue, the paper proposes 3 technical improvements. First, during training, the model only decodes voxels around the surface of the object, given by the ground truth mesh’s depth map. Second, the method adjusts the camera perspective to fit a cropped version of the voxel set in frame. Third, the method adds a TSDF and a TV loss to regularize the generated geometry.

**Compliance With Llm Reviewing Policy:**

Affirmed.

**Final Justification:**

The paper makes solid technical contributions in 3D generation, reducing the memory usage of existing approaches. Although the original submission contained many typos, I am happy to maintain my score as the authors promised to improve the writing in a revision.

**Key Questions For Authors:**

1. As mentioned in the weaknesses, I would appreciate some help understanding the “zooming” technique in Section 3.3. How is $(x_{min}^n, x_{max}^n, y_{min}^n,  y_{max}^n)$ calculated from $\mathcal{Z}_{crop}^p$?

**Limitations:**

Yes

**Strengths And Weaknesses:**

## Strengths
- The method appears to work well with very detailed geometries. Visually, the autoencoded surfaces are high resolution, yet free of aliasing.
- The method significantly reduces the memory used during training, with the ablation study in Table 3 showing that the “Zooming” technique can reduce memory usage by 11-29 GB.
- Figure 2 clearly illustrates the difference between the sparsification approaches of the proposed method and prior work.
- The paper is self-contained. A lot of work has been published on 3D generation with VAEs in recent years. The related work in section 2.2 summarizes the most relevant prior work very clearly.


## Weaknesses
- The improvement in detail seems very marginal compared to prior works like TripoSF. Quantitative metrics show that the performance of the proposed method and TripoSF are about the same.
- I had a difficult time understanding Section 3.3 and how it relates to dolly zoom. Typically a dolly zoom consists of changing the focal length and camera position at the same time so the the size of the subject stays the same. But it seems that the method only modifies the camera’s perspective matrix. I was also confused about why the camera frustum stays the same in Figure 2 (d), compared to Figure 2(a). I expected the frustum to change so that it would also be similar to Figure 2(c).
- I was surprised to see no video results. 360 videos would be good to show that the generated objects are free of artifacts and flickering.
- I think there is a typo in Eq. 4. The element in the second row and last column should be in terms of y, not in terms of x.


##  Other Typos
L120: “Conceptually illustration” -> “Conceptual illustration”
L367 “Conclusions & Limitaion"-> “Conclusions & Limitations

Overall I think the paper presents some solid efficiency improvements for 3D VAEs, but the presentation should be polished in a future revision.

---

> ### Author Rebuttal · Authors · 2026-03-31
>
> **Q: The improvement in detail seems very marginal compared to prior works like TripoSF. Quantitative metrics show that the performance of the proposed method and TripoSF is about the same.**
>
> A: It is indeed difficult for the Chamfer Distance metric to adequately reflect our method's superiority in capturing fine details and producing fewer holes compared to TripoSF. Furthermore, even within the Level 4 subset of the Dora benchmark, the data complexities are varied. Currently, there are no high-fidelity, high-quality datasets explicitly designed for such testing. Nevertheless, the qualitative results presented in the figures clearly demonstrate our method's advantages in capturing intricate details and preserving surface continuity.
>
> **Q: I had a difficult time understanding Section 3.3 and how it relates to dolly zoom. Typically a dolly zoom consists of changing the focal length and camera position at the same time so the the size of the subject stays the same. But it seems that the method only modifies the camera’s perspective matrix. I was also confused about why the camera frustum stays the same in Figure 2 (d), compared to Figure 2(a). I expected the frustum to change so that it would also be similar to Figure 2(c).**
>
> A: We thank the reviewer for pointing out this typo. We refer to our method as "adaptive zooming." We will correct the erroneous reference to "dolly zoom" in Line 238 in the revision. While we do not alter the near and far planes as depicted in Fig. 2(c), we do indeed adjust the focal length of the camera. We will add visual elements related to zooming in the updated Fig. 2(d) to make this clearer.
>
> **Q: I was surprised to see no video results. 360 videos would be good to show that the generated objects are free of artifacts and flickering.**
>
> A: We have added rendered 360° videos for several models in the [anonymous link](https://anonymous.4open.science/r/More_result-98BB/README.md). We hope this better illustrates the qualitative differences between our method and prior works.
>
> **Q: I think there is a typo in Eq. 4. The element in the second row and last column should be in terms of y, not in terms of x.**
>
> A: Thank you for pointing out this typo, we will fix it in the revision.
>
> **Q: Details about  Section 3.3.**
>
> A: We first randomly select a voxel to serve as a seed. Then, using K-Nearest Neighbors in the camera space, we find the $\alpha N$ voxels closest to this seed. The camera space coordinates of these voxels are used to calculate the initial $(x^{n}\_{min}, x^{n}\_{max},y^{n}\_{min},y^{n}\_{max})$. Since this bounding box contains greater than or equal to $\alpha N$ voxels, we subsequently use a binary search method to update our bounding box until the difference between the number of voxels inside the bounding box and the target number is less than 5%.

---

> > ### Author Rebuttal · Reviewer_jWX3 · 2026-03-31
> >
> > Thank you for the response.
> >
> > As most of my concerns were about the presentation of the paper rather than the technical contribution itself, I am happy that the authors promise to improve the writing. However, the contribution and improvements remain marginal over prior work like TripoSF. Weighing these factors, I am inclined to keep the same score.
> >
> > I will update my final review after discussing with the other reviewers.

---

> > > ### Author Response · Authors · 2026-04-02
> > >
> > > Dear Reviewer jWX3,
> > >
> > > Thank you for your continued engagement and for acknowledging that our rebuttal resolved your concerns.
> > >
> > > We are glad that our explanations clarified your questions regarding the presentation and technical details. As promised in our rebuttal, we are fully committed to polishing the writing and presentation in the final revision. Specifically, we will:
> > >
> > > 1) **Correct all typos:** This includes fixing the variable error in Eq. 4 and the spelling mistakes in the text.
> > >
> > > 2) **Clarify algorithm details:** We will remove the misleading reference to "dolly zoom" and accurately describe the exact mechanism and calculation of our "adaptive zooming" technique.
> > >
> > > 3) **Update figures and supplementary materials:** We will update Figure 2(d) to visually clarify the focal length adjustments and ensure the 360° rendered videos are properly referenced/hosted to better demonstrate our method's advantages in preserving fine details and avoiding artifacts.
> > >
> > > Thank you again for your time and the constructive feedback you have provided to improve our paper!
> > >
> > > Best regards,
> > > The Authors

---

### Official Review · Reviewer_2Qv6 · 2026-03-11

**Soundness:** 3
**Presentation:** 2
**Significance:** 3
**Originality:** 3
**Overall Recommendation:** 4
**Confidence:** 2

**Summary:**

Focusing" is a 3D Variational Auto-Encoder (VAE) designed to reconstruct high-fidelity 3D models at 1024^3 resolution with high efficiency.

The core innovation is view-consistent voxel carving performed in a structured latent space, which prunes voxels inconsistent with the rendered depth map before they reach the decoder.

This strategy, combined with an adaptive zooming mechanism that dynamically adjusts camera intrinsics to keep active voxels within a target range (8,192–15,360), significantly reduces VRAM usage to under 50GB and nearly doubles training throughput.

The model represents geometry using SparseFlex, which outputs Signed Distance Function (SDF) values, deformations, and interpolation weights for mesh extraction via Dual Marching Cubes

**Compliance With Llm Reviewing Policy:**

Affirmed.

**Key Questions For Authors:**

1. Feature Dimensionality: What is the specific channel dimension of the feature vector within each latent voxel before it is decoded into SparseFlex parameters?

2. Temporal Consistency: Since carving is view-dependent, how does the model ensure temporal or spatial consistency across different viewpoints during a single generative inference session?

3. Encoder Optimization: Are there plans to implement sparse operations within the PointNet encoder to overcome the global processing bottleneck and enable training at 1536^3 and beyond?

4. Thin Structure Mitigation: Could a hybrid representation (e.g., combining voxels with surfels or primitives) resolve the failure cases for sub-voxel thickness?

**Limitations:**

The model’s primary technical limitation is its spatial resolution constraint at the encoder stage; the requirement for global information prevent direct training at resolutions exceeding 1024^3.

Geometrically, the reliance on a structured voxel grid makes the system unable to recover extremely thin structures. Additionally, while rendering-based supervision avoids lossy SDF preprocessing, it provides weaker constraints than direct SDF supervision, occasionally leading to under-regularized latent voxels or small "holes" in the mesh.

Finally, the method's performance is strictly bound by the quality of input point cloud features and their associated normal vectors.

**Strengths And Weaknesses:**

Strengths:

Memory Efficiency: Enables 1024^3 resolution training on as little as 50GB of VRAM, a 37% reduction over non-carved baselines.

High-Resolution Fidelity: Outperforms state-of-the-art methods like TripoSF in geometric accuracy metrics, specifically Chamfer Distance (CD) and F-score.

Zero-Shot Generalization: A model trained at 1024^3 resolution can successfully perform 15363 reconstructions without further training.

Weaknesses:

Thin Structure Failure: The voxel-based representation inherently struggles to capture or reconstruct geometric elements thinner than a single voxel, such as fine hair.

Normal Vector Dependency: The final output quality is highly sensitive to the accuracy of the input point cloud’s normal vectors.

Encoder Bottleneck: Since the encoder still processes global information before carving, the method cannot yet be scaled directly to 15363 training resolution.

---

> ### Author Rebuttal · Authors · 2026-03-31
>
> **Q: Feature Dimensionality: What is the specific channel dimension of the feature vector within each latent voxel before it is decoded into SparseFlex parameters?**
>
> A: The initial feature dimension of each latent voxel in the latent space is 8. Within the 3D sparse decoder, this feature is expanded to a 64-dimensional hidden representation, and is subsequently mapped by the final linear projection layer into a final 53-dimensional vector.
>
> **Q: Temporal Consistency: Since carving is view-dependent, how does the model ensure temporal or spatial consistency across different viewpoints during a single generative inference session?**
>
> A: During inference, we do not employ a view-dependent approach. We use a chunk-based processing method to reduce the peak VRAM footprint when decoding the latent. This entirely avoids the need to stitch together meshes generated from different viewpoints.
>
> **Q: Encoder Optimization: Are there plans to implement sparse operations within the PointNet encoder to overcome the global processing bottleneck and enable training at $1536^3$ and beyond?**
>
> A: We thank you for the suggestion. We will actively explore ways to reduce the VRAM consumption of the PointNet encoder in our future work.
>
> **Q: Thin Structure Mitigation: Could a hybrid representation (e.g., combining voxels with surfels or primitives) resolve the failure cases for sub-voxel thickness?**
>
> A: We appreciate your suggestion and plan to experiment with a hybrid representation to address this in our future work.

---

> > ### Author Rebuttal · Reviewer_2Qv6 · 2026-04-02
> >
> > From all the reviews and rebuttals, the paper would improve by adding more details and explanations.

---

> > > ### Author Response · Authors · 2026-04-02
> > >
> > > Dear Reviewer 2Qv6,
> > >
> > > Thank you for your continued engagement and for confirming that our rebuttal has fully resolved your concerns. We sincerely appreciate your recognition of our work (Weak Accept) and the valuable feedback you provided throughout the review process.
> > >
> > > We will carefully polish the manuscript to ensure that the final version is comprehensive, clear, and easy for readers to reproduce. Thank you once again for your support and the time you have dedicated to evaluating our research!
> > >
> > > Best regards,
> > >
> > > The Authors

---

### Official Review · Reviewer_tF7n · 2026-03-14

**Soundness:** 3
**Presentation:** 3
**Significance:** 3
**Originality:** 2
**Overall Recommendation:** 4
**Confidence:** 2

**Summary:**

This paper proposes a method, Focusing, for efficient 3D VAE. The method follows a render-based training paradigm and introduces depth-driven voxel carving in the latent space, which activates only the voxels relevant to a given view before decoding. This reduces computation and VRAM usage while concentrating learning on locally important geometry. The paper also further proposes an adaptive zooming strategy to stabilize training and maintain an appropriate number of active voxels, along with simple regularizers such as sparse-voxel TV and a short TSDF warm-up. Experiments on reconstruction benchmarks show improved geometric accuracy and reducing memory usage.

**Compliance With Llm Reviewing Policy:**

Affirmed.

**Key Questions For Authors:**

1. Could the authors comment on when the proposed carving strategy helps the most, and whether there are cases where it brings limited benefit?

2. Missing qualitative ablations for carving thresholding, zoom schedule, etc, to show the performance gain from each design.

**Limitations:**

yes

**Strengths And Weaknesses:**

**Soundness**: The method appears technically reasonable. The paper proposes a depth-based voxel carving strategy in the structured latent space, together with adaptive zooming and a few simple regularizers, to improve the efficiency and detail quality of a render-supervised 3D VAE. The empirical results on Dora benchmark reconstruction suggest modest but consistent improvements over TripoSF in Chamfer Distance and F-score, and the ablation on training cost also supports the efficiency claim.

**Presentation**: The paper is generally clear and reasonably well organized. However, I don't have strong expertise in this area, so I can't fully judge whether the paper provides sufficient information.

**Significance & Originality**: The paper studies a topic of efficient 3D VAE, which is a critical task for many downstream tasks, such as 3D generation. The benefit is essential, especially if reducing VRAM indeed helps scale training to 1024³ resolution more easily. Although my impression is that the contribution is more of a solid systems/method refinement than a major conceptual breakthrough. But overall, this feels meaningful enough for acceptance, given the improvement it achieves.

---

> ### Author Rebuttal · Authors · 2026-03-31
>
> **Q: Could the authors comment on when the proposed carving strategy helps the most, and whether there are cases where it brings limited benefit?**
>
> A: For complex objects or high-resolution training, carving is an extremely effective method to reduce VRAM requirements. However, for low-resolution training (e.g., $256^3$ resolution), the benefits of carving are limited. At such low resolutions, fine details are difficult to represent, and baseline VRAM usage is already low, making the performance difference of using carving marginal.
>
> **Q: Missing qualitative ablations for carving thresholding, zoom schedule, etc, to show the performance gain from each design.**
>
> A: We have provided qualitative ablations for the carving thresholding in Figure 9 of the supplementary materials.

---

> > ### Author Rebuttal · Reviewer_tF7n · 2026-04-03
> >
> > After reading all the reviews and the rebuttal, I tend to keep my original score of weak accept.
> >
> > One additional minor concern is that Figure 9 is not referenced in the main text. Figures should be properly cited and discussed to ensure clarity. I also suggest that the authors further improve the overall presentation for better readability.

---

### Official Review · Reviewer_aWGh · 2026-03-18

**Soundness:** 2
**Presentation:** 2
**Significance:** 3
**Originality:** 3
**Overall Recommendation:** 4
**Confidence:** 3

**Summary:**

This paper proposes Focusing, which introduces novel methods to improve the training efficiency of 3D VAEs. The paper introduces two key methods: (i) a depth-driven, view-consistent voxel carving method in the latent space to prune and maintain only visible parts, and (ii) adaptive zoom-in by dynamically adjusting the camera intrinsics to keep the number of voxels within a certain range preventing memory fluctuation. Also, additional regularizations such as $L_{occ}$, $L_{tsdf}$, and $L_{tv}$ is introduced for improved performance. Evaluation on Dora Benchmarks validates the performance improvement of Focusing over TripoSF. Additional qualitative results show the potential of Focusing to be used in image-to-3D generation pipelines.

**Compliance With Llm Reviewing Policy:**

Affirmed.

**Final Justification:**

The rebuttal of the authors has clarified most of the ambiguous parts of the paper and addressed most of my concerns. I believe the efficiency and effectiveness of the proposed method are the main contributions of the paper, and I strongly recommend that the authors improve the overall presentation of the paper and include important details, such as the camera viewpoint sampling strategy. I will raise my rating to weak accept.

**Key Questions For Authors:**

Most of my questions are specified in my weaknesses section. I would appreciate it if the authors could address my questions.
One additional question I would like to ask is:

1. For the $L_{occ}$, I have noticed that the output of the self-pruning block, $O_{pred}$, is not used during training. Does using this  $O_{pred}$ only at inference time not harm the robustness of the inference pipeline? Have you tried teacher-forcing type of approaches or using this  $O_{pred}$ after sufficient training steps?

With my questions being addressed, I am eager to raise my score.

**Limitations:**

yes.

**Strengths And Weaknesses:**

### Strengths

---

**S1. Introduced methods are intuitive.** The introduced key method which performs voxel carving directly in latent space before decoding is simple and intuitive. The proposed method improves previous approaches of adjusting near and far planes after decoding. Also the dynamic zoom-in method by controlling the camera intrinsics to maintain similar number of voxels is also reasonable.

**S2. Improves efficiency of training 3D VAEs.** The proposed method largely improves training efficiency verified in Table 3. The table verifies that applying voxel carving largely improves efficiency (OOM $\rightarrow$ 78GB), and adaptive zoom-in further largely improves both training speed and peak gpu memory training. This strength largely improves efficiency during training, which can enable more users to train 3D VAEs within practical computations.

**S3. Consistent reconstruction improvement.** While not being significant, Focusing improves both chamfer distance and F-score over TripoSF at both 512 and 1024 resolutions across all geometry detail levels of the Dora Benchmark.

**S4. Generalization to higher resolution decoding.** The authors showcase that a model trained at 1024³ can produce reasonable reconstructions at 1536³ without additional training. This verifies that training with local geometry enables generalization to even higher resolutions unseen during training at inference time.

### Weaknesses

---

**W1. Train-test mismatch for the decoder is not analyzed.** During training, the decoder only processes 8,192–15,360 voxels (thin surface shells from one viewpoint, possibly zoomed in). During inference, it must decode the complete voxel set, which could be an order of magnitude larger and includes interior, back-facing, and occluded regions never seen during training. The paper relies on the implicit argument that local attention makes this safe, but provides no ablation or analysis. I would like to find additional analysis such as comparing reconstruction quality when decoding full sets versus stitching per-view crops at inference.

**W2. The main goal of this paper is ambiguous in the current form of presentation.** The current presentation of the paper introduces a new methods to improve the efficiency and effectiveness of 3D VAEs. In addition, the paper also mentions rectified flow based image-to-3D generation. While this generative pipeline is also emphasized, there are no detailed analysis or evaluation of this pipeline. The results in Figure 6 only show qualitative results, which is difficult to understand general performance of replacing existing 3D VAEs with the proposed Focusing in generative pipelines. The paper should significantly increase the portion of analysis in generative pipelines or specifically constrain their analysis in reconstruction quality of 3D VAEs.

**W3. TSDF computation is a bit contradictory.** The paper's central motivation is avoiding lossy watertight conversion, yet $L_{tsdf}$ requires computing signed distances from raw meshes.

**W4. Camera viewpoint sampling strategy is undocumented.** The paper does not specify any details of how camera viewpoints are sampled during training. This is a more consequential omission here than in standard 3D VAE papers because the viewpoint distribution directly determines which voxels receive supervision via the carving mechanism. If certain surface regions are consistently underrepresented in the viewpoint distribution, their latent codes would be poorly trained. The interaction between viewpoint sampling and adaptive zooming further complicates this — the joint distribution of views and zoom levels determines the effective training distribution over the latent space.

**W5. Inference cost is unreported.** The paper emphasizes training efficiency but provides no inference-time VRAM usage, latency, or voxel count statistics. Since the efficiency argument is training-specific (carving and zooming are disabled at inference), the reader cannot assess whether full decoding at inference is practical, particularly at $1024^3$ or $1536^3$ resolution.

---

> ### Author Rebuttal · Authors · 2026-03-31
>
> **Q: W1. Train-test mismatch for the decoder is not analyzed.**
>
> A: Since our cropping is view-based rather than block-based, stitching per-view crops is quite difficult. As shown in Fig. 3, view-based cropping results in jagged structures at the boundaries of the generated mesh. Although this noise is invisible during the training phase, and we avoid this issue during inference by using the full volume of voxels as input, stitching per-view crops would require us to repair these jagged connections during inference.
>
> To validate the train-test mismatch for the decoder, we designed a new experiment. We randomly selected 100 assets from the benchmark, rendering 10 views for each. We compared the error between the mesh generated using the full voxel set and the sub-meshes generated from the views with L1 depth loss. The results show that the depth error is on the scale of $10^{-4}$, which is practically negligible.
>
> **Q: W2. The main goal of this paper is ambiguous in the current form of presentation.**
>
> A: We provide the image-to-3D generation results to verify that the latent space of our VAE complies with the KL divergence constraint and can be applied to 3D AIGC tasks. We do not claim that our rectified flow model is state-of-the-art, as we primarily focus on introducing a new training strategy for 3D VAEs. We will further clarify the rationale for including these 3D AIGC results in the revision.
>
> **Q: W3. TSDF computation is a bit contradictory. The paper's central motivation is avoiding lossy watertight conversion, yet requires computing signed distances from raw meshes.**
>
> A: We use zero weights to initialize the parameters of the output layer in our network. This causes the initial SDF values output by our VAE to be exactly zero, making it impossible to extract a mesh for gradient backpropagation. In TRELLIS, the authors subtract a bias value from the SDF values of the sparse voxels and combine them with an all-zero background to obtain a dense 3D volume for mesh extraction. In our framework, we do not perform this densification operation. Therefore, we require an additional loss to help optimize the initial network weights. We utilize the TSDF loss for this purpose, but it is applied only during the very early stages of training.
>
> **Q: W4. Camera viewpoint sampling strategy is undocumented.**
>
> A: We thank the reviewer for highlighting this critical omission. We completely agree that in 3D VAEs, the joint distribution of viewpoints and zoom levels directly determines the supervision quality of the carving mechanism. To completely eliminate the risk of under-representing any surface region, we designed a "Hybrid Uniform Sampling with Stratified Zooming" strategy for training data preparation (300 views in total), detailed as follows:
>
> 1. **Deterministic Global Anchors (2/3 views, fixed scale):** We generate 200 views using a Fibonacci spherical lattice, which mathematically guarantees a nearly perfect isotropic uniform distribution, strictly avoiding polar bias. Adaptive zooming is disabled for these views: 100 use a fixed far scale (Radius 2.0, FOV $40^{\circ}$), and 100 use a medium scale (Radius 1.5, FOV $40^{\circ}$). They serve as zero-variance "structural anchors" for the carving mechanism, ensuring no macro-geometry becomes a supervision blind spot.
> 2. **Stochastic Multi-scale Augmentation (1/3 views, random zoom):** The remaining 100 views are generated using a Hammersley sequence with a random global phase offset. For these views, aggressive adaptive zooming is enabled (Radius randomly sampled $\in [0.8, 2.2]$, FOV $\in [20^{\circ}, 70^{\circ}]$). This acts as stochastic data augmentation, forcing the latent code to learn multi-scale, high-frequency local details and continuous viewpoint interpolation.
>
> **Q: W5. Inference cost is unreported.**
>
> A: Our inference cost is similar to that of TripoSF. Since gradients do not need to be computed during inference, VRAM consumption is generally not a severe bottleneck. By decoding in chunks during the inference phase, we can effectively control peak VRAM usage. A similar chunking operation is also utilized in the released inference code for TripoSF (see line 295 in https://github.com/VAST-AI-Research/TripoSF/blob/main/triposf/models/triposf_vae/decoder.py).
>
> **Q: The use of $O_{pred}$.**
>
> A: Based on our empirical training results, applying self-pruning during training is actually harmful. This is because differentiable rendering struggles to distinguish between artifacts caused by self-pruning and those caused by SDF fluctuations. We appreciate the reviewer's suggestion; utilizing $O_{pred}$ after it has stabilized could indeed be a viable approach. However, determining the specific strategy would require additional experimentation, and we plan to explore incorporating this mechanism in our future work.

---

> > ### Author Rebuttal · Reviewer_aWGh · 2026-04-02
> >
> > The rebuttal of the authors has clarified most of the ambiguous parts of the paper and addressed most of my concerns. I believe the efficiency and effectiveness of the proposed method are the main contributions of the paper, and I strongly recommend that the authors improve the overall presentation of the paper and include important details, such as the camera viewpoint sampling strategy. I will raise my rating to weak accept.

---

> > > ### Author Response · Authors · 2026-04-02
> > >
> > > Dear Reviewer aWGh,
> > >
> > > Thank you very much for taking the time to read our rebuttal and for acknowledging our clarifications. We sincerely appreciate your decision to raise your score to a Weak Accept, as well as all the constructive feedback you have provided throughout the review process to improve the quality of our paper.
> > >
> > > We completely agree with your final suggestions. In the upcoming camera-ready version, we are committed to strictly following your recommendations to revise and refine the paper. Specifically, we will:
> > >
> > > 1) **Improve the Overall Presentation:** We will refine the structure and phrasing to ensure that the core contribution of our paper—the efficiency and effectiveness of the proposed 3D VAE training method—is highlighted more clearly. We will also clarify the role of the 3D AIGC (Image-to-3D) generation results as supplementary validation to avoid any ambiguity regarding the main goal of the paper.
> > >
> > > 2) **Include Camera Viewpoint Sampling Details:** We will add a detailed and complete description of the "Hybrid Uniform Sampling with Stratified Zooming" strategy mentioned in our rebuttal to the main text or appendix.
> > >
> > > 3) **Integrate Key Analyses from the Rebuttal:** We will incorporate other important discussions from the rebuttal into the revised manuscript. This includes the explicit motivation behind the TSDF computation, the clarification regarding inference costs, and the discussion on the potential future exploration of the self-pruning module.
> > >
> > > Thank you again for your recognition and support of our work!
> > >
> > > Best regards,
> > >
> > > The Authors

---

### Decision · Program_Chairs · 2026-04-30

**Decision:**

Accept (regular)

**Comment:**

This paper proposes an efficient 3D VAE based on view-consistent voxel carving and adaptive zooming, to improve the training efficiency while maintaining high-fidelity reconstruction. Reviewers found the method technically sound and appreciated its practical contribution in reducing VRAM usage and enabling higher-resolution training.

While the novelty is primarily incremental, the empirical improvements and efficiency gains were viewed as meaningful. The AC therefore recommends acceptance. At the same time, the authors should revise the manuscript carefully to improve the presentation and include more implementation details. In particular, important details such as the camera viewpoint sampling strategy and training / inference details should be described more clearly in the final version.